# Conformational and oligomeric states of SPOP from small-angle X-ray scattering and molecular dynamics simulations

F Emil Thomasen[1], Matthew J Cuneo[2], Tanja Mittag[2], Kresten Lindorff-Larsen[1]*

[1]Linderstrøm-Lang Centre for Protein Science, Department of Biology, University of Copenhagen, Copenhagen, Denmark; [2]Department of Structural Biology, St. Jude Children's Research Hospital, Memphis, United States

**Abstract** Speckle-type POZ protein (SPOP) is a substrate adaptor in the ubiquitin proteasome system, and plays important roles in cell-cycle control, development, and cancer pathogenesis. SPOP forms linear higher-order oligomers following an isodesmic self-association model. Oligomerization is essential for SPOP's multivalent interactions with substrates, which facilitate phase separation and localization to biomolecular condensates. Structural characterization of SPOP in its oligomeric state and in solution is, however, challenging due to the inherent conformational and compositional heterogeneity of the oligomeric species. Here, we develop an approach to simultaneously and self-consistently characterize the conformational ensemble and the distribution of oligomeric states of SPOP by combining small-angle X-ray scattering (SAXS) and molecular dynamics (MD) simulations. We build initial conformational ensembles of SPOP oligomers using coarse-grained molecular dynamics simulations, and use a Bayesian/maximum entropy approach to refine the ensembles, along with the distribution of oligomeric states, against a concentration series of SAXS experiments. Our results suggest that SPOP oligomers behave as rigid, helical structures in solution, and that a flexible linker region allows SPOP's substrate-binding domains to extend away from the core of the oligomers. Additionally, our results are in good agreement with previous characterization of the isodesmic self-association of SPOP. In the future, the approach presented here can be extended to other systems to simultaneously characterize structural heterogeneity and self-assembly.

**\*For correspondence:** lindorff@bio.ku.dk

## Editor's evaluation

In this important paper, the authors have developed an approach for simultaneously optimizing the conformational ensemble and degrees of oligomerization, and this has been tested by applying it to a specific protein (SPOP). Comparison of the quality of fits with different models also provides valuable insights into structural features important to the assembly of oligomers. The approach, presented with compelling experimental support, is potentially applicable to other systems as well.

## Introduction

Protein self-association is fundamental for many processes in biology (*Ali and Imperiali, 2005*; *Marsh and Teichmann, 2015*), and it has been estimated that around half of all proteins form dimers or higher-order complexes (*Lynch, 2012*). One such protein is Speckle-type POZ protein (SPOP), a substrate adaptor in the ubiquitin proteasome system, which recruits substrates for the Cullin3-RING ubiquitin ligase (CRL3) (*Hernández-Muñoz et al., 2005*; *Kent et al., 2006*; *Kwon et al., 2006*). SPOP targets a range of substrates for degradation, including proteins involved in hormonal signalling,

**Figure 1.** SPOP forms higher-order oligomers through isodesmic self-association. (**a**) The SPOP BTB-BTB homodimer forms with nanomolar affinity, and is the unit of higher-order oligomerization through BACK-BACK homodimerization. Higher-order SPOP oligomerization follows an isodesmic model, where the equilibrium between oligomer $i$ and $i$+1 is described by a single equilibrium constant, $K_{D,isodesmic}$, which is independent of oligomer size. (**b**) Crystal structures of homodimers of the BACK (left, PDB: 4HS2) and MATH-BTB (right, PBD: 3HQI) domains of SPOP. Below, the structure of a SPOP[28–359] dimer constructed based on crystal structures. The cartoon model is overlaid with the coarse-grained representation used in the Martini simulations. The BACK domains of the two neighbouring subunits in the oligomer are also shown (without Martini bead overlay). (**c**) Left: Populations of SPOP oligomers given by the isodesmic model with $K_{D,isodesmic}$=1.6 µM, determined from CG-MALS, for the protein concentrations used in our SAXS experiments. Note the logarithmic scale. Right: Relative contribution of each oligomer to the average SAXS signal given by the populations in left panel. (**d**) Structure of a SPOP[28–359] 60-mer constructed based on structures in panel b. MATH domains are coloured orange and BTB/BACK domains are coloured blue in all panels.

The online version of this article includes the following figure supplement(s) for figure 1:

**Figure supplement 1.** Fit of isodesmic model to CG-MALS.

epigenetic modification, and cell-cycle control, such as the androgen receptor (AR) (*An et al., 2014*) and death-associated protein 6 (DAXX) (*Kwon et al., 2006*; *Cuneo and Mittag, 2019*). SPOP is thus an important regulator of cellular signalling, and mutations in SPOP are associated with a variety of cancers (*Le Gallo et al., 2012*; *Kim et al., 2013*; *Cuneo and Mittag, 2019*).

The 374-residue SPOP monomer consists of three domains. From N- to C-terminus, these are the MATH domain (i.e., the meprin and TRAF-C homology domain), the BTB domain (i.e. the broad-complex, tramtrack, and bric-a-brac domain), and the BACK domain (i.e. the BTB and C-terminal Kelch domain). MATH is the substrate binding domain, while the BTB domain mediates interaction with CRL3 (*Zhuang et al., 2009*; *Bosu and Kipreos, 2008*). Both the BTB and BACK domains can homodimerize, resulting in the formation of polydisperse, linear higher-order SPOP oligomers with alternating BTB-BTB and BACK-BACK interfaces (*Errington et al., 2012*; *van Geersdaele et al., 2013*; *Marzahn et al., 2016*). The BTB-mediated dimer is formed with nanomolar affinity, and this dimer acts as the unit of higher-order oligomerization, which occurs through micromolar affinity BACK dimerization. Thus, only even-numbered SPOP oligomers are substantially populated (*Marzahn et al., 2016*; *Figure 1*).

Chemical crosslinking experiments have shown that SPOP oligomers form inside cells (*Marzahn et al., 2016*), and analysis of SPOP homologues shows sequence co-variation across both the BTB-BTB and BACK-BACK interfaces (*Bouchard et al., 2018*), together suggesting that self-association has physiological relevance. By presenting multiple MATH domains for substrate binding, SPOP oligomers can simultaneously bind to multiple low-affinity binding motifs in a single substrate, resulting

in an overall increased affinity through avidity effects (*Pierce et al., 2016*). The longer lifetimes of these complexes enable effective polyubiquitination (*Pierce et al., 2016*). This suggests that tuning SPOP's oligomerization state could act as a mechanism to regulate substrate degradation (*Errington et al., 2012*). SPOP oligomerization is also involved in phase separation. SPOP localizes to nuclear speckles in cells (*Marzahn et al., 2016*), and upon overexpression of certain substrates, SPOP and substrate co-localize to condensates which recruit CRL3 and display active substrate ubiquitination (*Bouchard et al., 2018*). This process requires both SPOP oligomerization and substrate binding (*Marzahn et al., 2016*; *Bouchard et al., 2018*), and it has been proposed that SPOP oligomers function as scaffolds that enable binding of substrates both within and between oligomers, resulting in filament-formation at low substrate concentrations and condensate formation at higher substrate concentrations (*Bouchard et al., 2018*; *Schmit et al., 2020*).

The higher-order self-association of SPOP follows the isodesmic model (*Marzahn et al., 2016*), in which the equilibrium between oligomer $i$ and $i+1$ is described by a single equilibrium constant independently of oligomer size (*Oosawa and Kasai, 1962*). In the case of SPOP, the BTB-mediated dimer acts as the protomer of higher-order self-association, and the isodesmic $K_D$ thus describes BACK-BACK self-association. The isodesmic model can be used to calculate the equilibrium concentration of every oligomeric species as a function of the total protomer concentration (*Figure 1a and c*). For SPOP, a low micromolar isodesmic $K_D$ has been determined from composition gradient multi-angle light scattering experiments (CG-MALS; *Marzahn et al., 2016*). While these insights describe the heterogeneity in oligomer sizes, the conformational heterogeneity of the higher-order oligomers has not been characterized. Previous work revealed that constitutive SPOP dimers, created via deletion of the BACK domain, have considerable conformational heterogeneity in the position of their MATH domains. The MATH domains are seen docked onto the BTB dimer in the structure, but small-angle X-ray scattering (SAXS) experiments showed that they could undock from the BTB domains, enabled by a long flexible linker (*Zhuang et al., 2009*). This may enable binding of multivalent substrates with different spacing between SPOP binding motifs. Whether this conformational flexibility also exists in higher-order SPOP oligomers is unclear.

Here, we aimed to determine simultaneously both the distribution of oligomeric states of SPOP and the conformational ensemble of each SPOP oligomer by combining SAXS experiments and MD simulations. SAXS can provide low-resolution information on protein structure in solution, but reports on an ensemble average, which in the case of SPOP is both an average over different oligomeric states and the structural heterogeneity of each oligomeric state. Therefore, SAXS experiments are often combined with MD simulations to provide a full structural model of the system (*Thomasen and Lindorff-Larsen, 2022*). In the case of polydisperse systems, it is sometimes possible to deconvolute the information into contributions from a small number of individual species and analyse these individually (*Herranz-Trillo et al., 2017*; *Meisburger et al., 2021*). We took a different approach and aimed to explicitly model every relevant configuration of SPOP in its range of oligomeric states along with the associated thermodynamic weight of each configuration. We collected SAXS data on SPOP at a range of protein concentrations and constructed initial conformational ensembles of every substantially populated oligomeric state using coarse-grained MD simulations. We then developed an approach to simultaneously and self-consistently optimize the distribution of oligomeric states, given by the isodesmic model (*Oosawa and Kasai, 1962*; *Shemesh et al., 2021*), and refine the conformational ensemble of each oligomer against the SAXS data using Bayesian/maximum entropy (BME) reweighting (*Bottaro et al., 2020*; *Figure 2*). Our results show that SPOP forms rigid, helical oligomers in solution, and that the linker connecting the MATH and BTB domains is likely flexible, allowing for repositioning of the MATH domains during substrate binding. Our results also provide further evidence that SPOP self-association follows the isodesmic model, and we find an isodesmic $K_D$ in good agreement with the previously determined value (*Marzahn et al., 2016*). Using SAXS experiments of a cancer variant of SPOP we also show how our approach can be used to determine changes in the level of self-association.

## Results

We collected a concentration series of SAXS data on a previously used truncated version of SPOP, SPOP[28–359] (full length is 374 residues), with total protein concentrations ranging from 5 to 40 μM. In order to build structural models to refine against the SAXS data, we first needed to decide which oligomeric

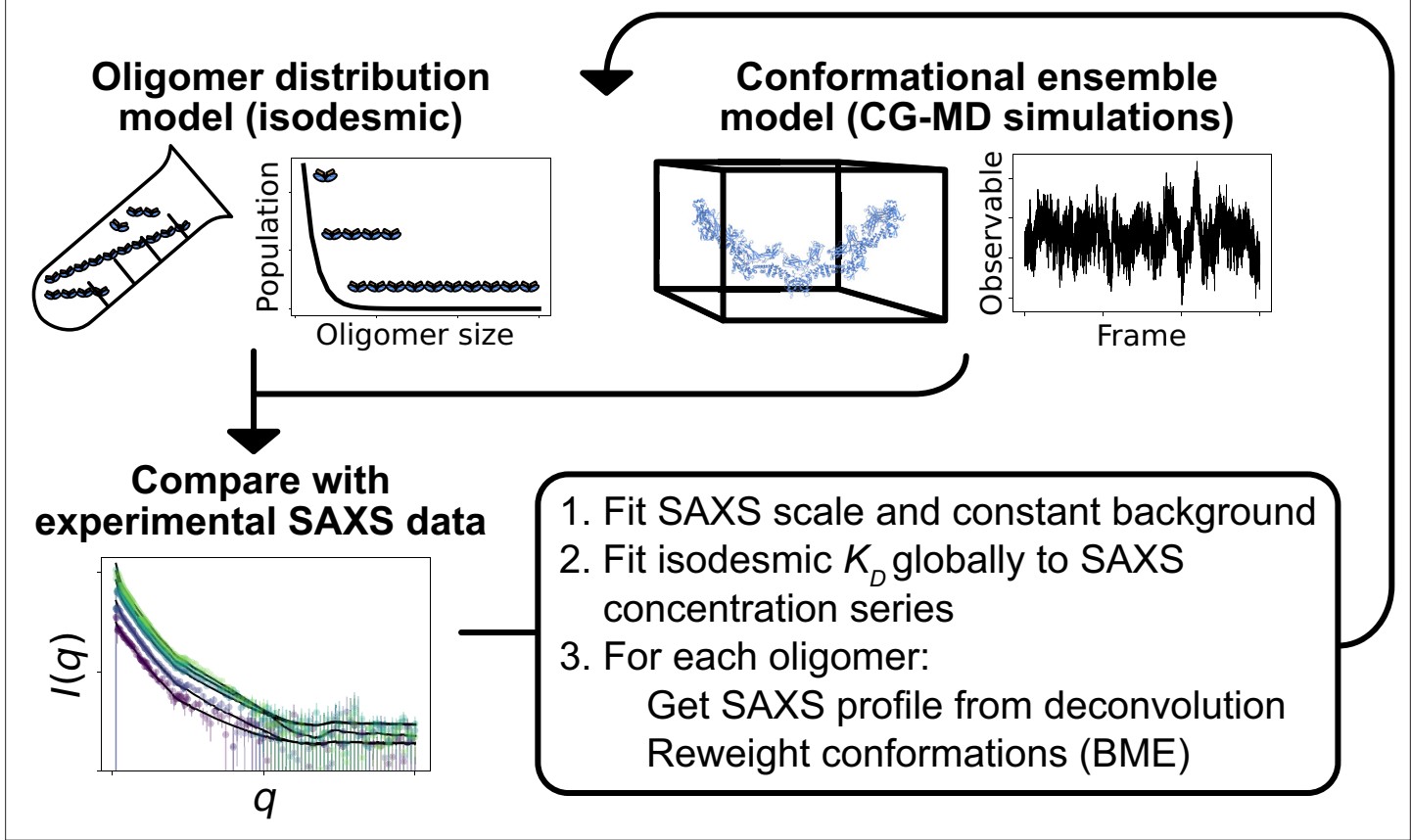

**Figure 2.** Overview of the self-consistent approach used to fit conformational ensembles of SPOP oligomers to SAXS data. Small-angle X-ray scattering (SAXS) data on SPOP represents an average over a range of oligomeric species present in solution. Here, the distribution of oligomeric species and the conformational ensemble of each oligomer were self-consistently fitted to a concentration series of SAXS data by iteratively fitting the scale and constant background of the SAXS data and the isodesmic $K_D$, followed by reweighting of the conformational ensemble of each oligomer.

species to include in our modelling. To this aim, we used the isodesmic self-association model, which has previously been shown to describe SPOP oligomerization well (***Marzahn et al., 2016***). We fitted previously measured CG-MALS data (***Marzahn et al., 2016***) to obtain an isodesmic $K_D$ of 1.6±0.3 μM (***Figure 1—figure supplement 1***). Based on the isodesmic model fitted to the CG-MALS data, the population of oligomers larger than ~30-mers should be very low at the concentration range used in our SAXS experiments. As scattering intensity is proportional to particle size squared, larger oligomers, however, make a considerable contribution to the SAXS signal despite their low concentrations (***Figure 1c***). Given the concentrations from the isodesmic model and taking into account the increased scattering of larger oligomers, we decided that constructing models of oligomers up to the 60-mer should be sufficient to capture all substantial contributions to the SAXS data.

There are no crystal structures of SPOP[28–359] available, so we constructed a model of the SPOP[28–359] BTB-dimer using the crystal structure of the isolated BACK domain (4HS2) (***van Geersdaele et al., 2013***) and the crystal structure of a truncated construct containing only the MATH and BTB domains (3HQI) (***Zhuang et al., 2009***; ***Figure 1b***). We used this model of the BTB-dimer to construct SPOP[28–359] oligomers, which we used as starting structures for MD simulations. We ran 60 μs MD simulations of oligomers ranging from the dimer to the dodecamer; we used a coarse-grained representation of SPOP modelled using the Martini 3 force field (***Souza et al., 2021***) further modified by increasing protein-water interactions by 6% (***Thomasen et al., 2022***). It would be computationally prohibitive to run simulations of large oligomers up to the 60-mer. Instead, we relied on the assumption that the dodecamer behaves similarly to a segment of an arbitrarily long oligomer, and constructed conformational ensembles of oligomers up to the 60-mer by joining together conformers from the simulations of the dodecamer at the BACK-BACK interface (***Figure 1d***).

We calculated SAXS intensities from our conformational ensembles and, given the relative population of each oligomer from the isodesmic model with $K_D$=1.6 µM, determined from CG-MALS, we calculated SAXS profiles averaged over all the oligomeric species. We found that the SAXS data calculated in this way from the ensembles generated by MD simulations convoluted with the isodesmic model were in good agreement with the experimental SAXS data, giving a reduced $\chi^2$ to the concentration series of SAXS data ($\chi^2_{r,\text{global}}$) of 1.53 (*Figure 3*). Despite the overall good agreement, the residuals revealed some systematic deviations to the experimental SAXS profiles. These deviations could potentially arise from inaccuracies in the distribution of oligomeric states given by the isodesmic model, from inaccuracies in the modelled conformational ensembles, or from both. As a first step, we wanted to see if we could eliminate the deviations by only tuning the distribution of oligomeric states. We globally optimized the $K_D$ of the isodesmic model against the concentration series of SAXS data, which gave $K_D$=0.9±0.4 µM, in good agreement with $K_D$=1.6±0.3 µM determined from CG-MALS, and resulted in a $\chi^2_{r,\text{global}}$ of 1.24 to the SAXS data (*Figure 3*). However, this did still not fully eliminate the systematic deviations from the experimental SAXS profiles.

To improve the agreement with the experimental SAXS data further, we aimed to simultaneously refine the conformational ensemble of each oligomer and optimize the distribution of oligomeric states. We developed a self-consistent optimization scheme, in which the isodesmic $K_D$ is optimized globally to the entire concentration series of SAXS data followed by reweighting of the conformations of each oligomer against a SAXS profile deconvoluted from the experimental SAXS data (*Figure 2*; see Methods section for details). To reweight the ensembles, we used BME reweighting, in which the population weights of the conformational ensemble are minimally perturbed with respect to the prior ensemble (generated by the MD simulations) to improve the agreement with a set of experimental data (*Bottaro et al., 2020*). This approach resulted in excellent agreement with the experimental SAXS data, giving a $\chi^2_{r,\text{global}}$ of 0.69, while only small deviations remained (*Figure 3*). The isodesmic $K_D$ was fitted to 1.3±0.5 µM, and thus also remained in good agreement with the previously determined value (*Marzahn et al., 2016*). To validate our approach and to examine the possibility of overfitting, we left out one SAXS profile recorded with 15 µM protein from the optimization. The optimized $K_D$ and ensemble weights did not substantially affect the fit to this SAXS profile, suggesting that we had avoided overfitting (*Figure 3—figure supplement 1*). These results show that SAXS data on SPOP can be explained well by conformational ensembles of linear oligomers with populations given by the isodesmic model, and thus provide further evidence that SPOP self-association follows a simple isodesmic mechanism (*Marzahn et al., 2016*).

The previously published CG-MALS data on SPOP clearly precludes a simple dimer–tetramer, dimer–hexamer, dimer–octamer, or dimer–decamer equilibrium in favor of an isodesmic self-association model (*Marzahn et al., 2016*). To determine whether the SAXS data also favors the isodesmic model, we used our conformational ensembles to examine whether monodisperse oligomers, ranging in size between an octamer and 60-mer, as well as corresponding dimer-oligomer equilibria, would be compatible with the SAXS concentration series (*Figure 3—figure supplement 2*). For each dimer–oligomer equilibrium, we fitted the $K_D$ globally to the SAXS data. Thus, the isodesmic model and dimer–oligomer models are of comparable complexity, with only a single free parameter. The results show that the SAXS concentration series is in better agreement with an isodesmic distribution of oligomers than with any of the tested single oligomers or dimer–oligomer equilibria.

We also wished to examine whether the conformational ensembles of SPOP generated by the MD simulations described the SAXS data better than static structures. As a first comparison, we calculated SAXS profiles from the initial SPOP oligomer structures constructed based on crystal structures. To make the results comparable with our optimized ensembles, we fitted the isodesmic $K_D$ to the SAXS data for the static structures. This resulted in a worse agreement with the SAXS data ($\chi^2_{r,\text{global}}$=4.03 with $K_D$=0.43 µM) than what we obtained using the ensembles both before and after reweighting. We also investigated the agreement with the SAXS data for individual structures drawn from the ensembles of the oligomers, again fitting the isodesmic $K_D$ for each set of structures. We found that some of the single structures from the ensembles could fit the SAXS data as well as the entire ensembles before reweighting, but no set of static structures fit the SAXS data as well as the reweighted ensembles (*Figure 3—figure supplement 3*). This result highlights that, while an ensemble of multiple conformers is likely necessary to produce the best agreement with the SAXS data, SPOP oligomers have a relatively rigid structure overall, allowing for reasonable agreement with the SAXS data without

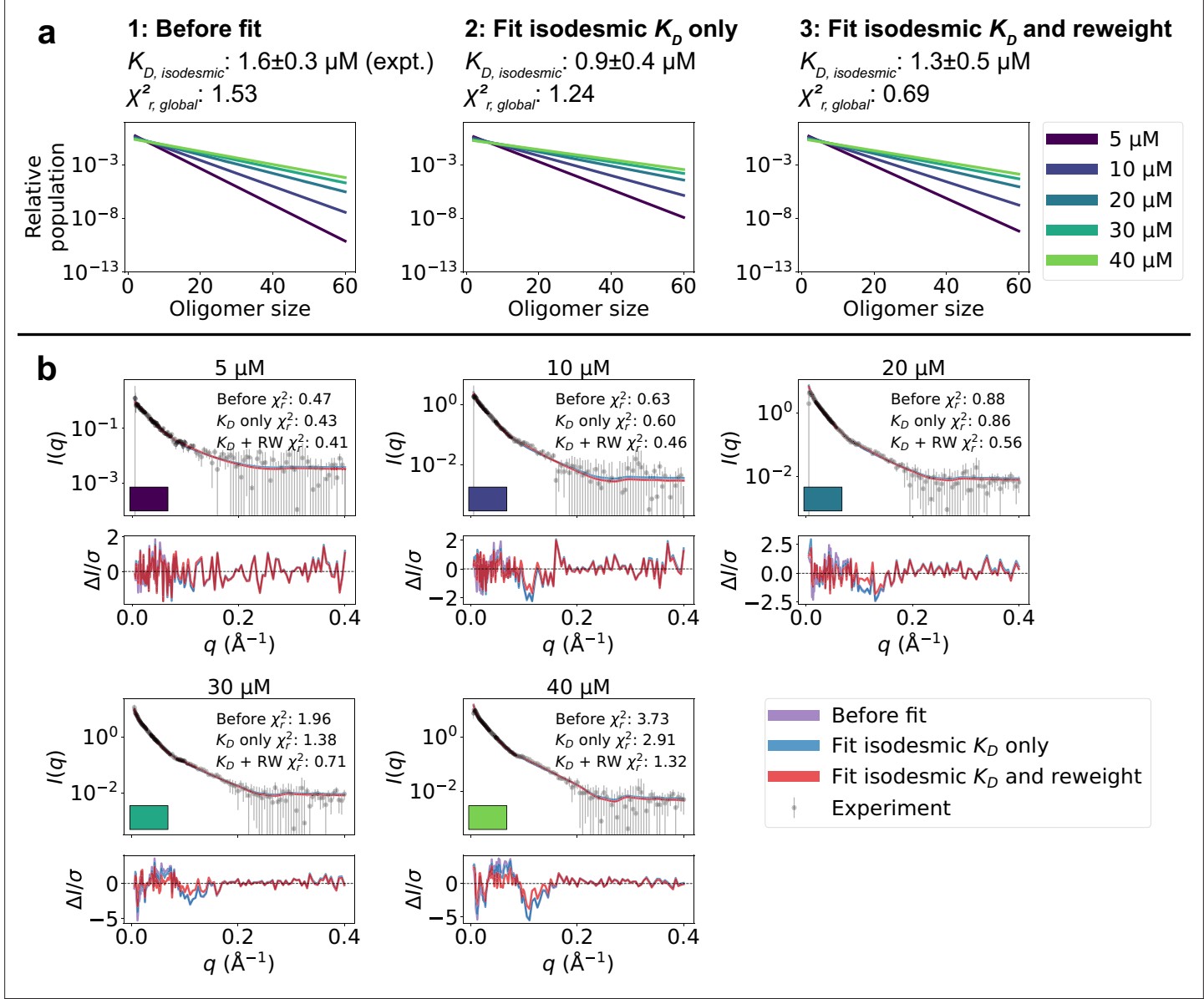

**Figure 3.** Refining oligomer populations and conformational ensembles against SAXS data. (**a**) Relative populations of oligomers for the protein concentrations used in SAXS experiments. Note the logarithmic scale. Populations are given by the isodesmic model with the $K_D$ noted above the plot, which is either (1) previously determined by CG-MALS or (2–3) fitted globally to the SAXS data in panel b. $\chi^2_{r,\mathrm{global}}$ quantifies the agreement with SAXS data in panel b for the three scenarios. (**b**) Agreement between experimental SAXS data and averaged SAXS data calculated from conformational ensembles of SPOP oligomers with populations given by the isodesmic model (as shown in panel a). SAXS profiles are shown for three different scenarios: (1) calculated from the conformational ensembles generated by MD simulations with the isodesmic $K_D$ previously determined with CG-MALS, (2) calculated from the conformational ensembles generated by MD simulations with the isodesmic $K_D$ fitted to the SAXS data, and (3) calculated from conformational ensembles refined against the SAXS data using Bayesian/MaxEnt reweighting, and with the isodesmic $K_D$ self-consistently fitted to the SAXS data. Error-normalized residuals are shown below the SAXS profiles and $\chi^2_r$ to each SAXS profile is shown on the plot.

The online version of this article includes the following figure supplement(s) for figure 3:

**Figure supplement 1.** Selection of $\phi_{\mathrm{eff}}$ and model validation.

**Figure supplement 2.** Agreement with SAXS for other self-association models.

**Figure supplement 3.** Comparison of static structures and ensembles.

**Figure supplement 4.** Agreement with CG-MALS for isodesmic model fitted to SAXS.

**Figure supplement 5.** Determining the error of the fitted isodesmic $K_D$ before reweighting.

**Figure supplement 6.** Determining the error of the fitted isodesmic $K_D$ after reweighting.

*Figure 3 continued on next page*

eLife Research article

Structural Biology and Molecular Biophysics

*Figure 3 continued*

**Figure supplement 7.** Fit to SAXS data for SPOP R221C.

**Figure supplement 8.** Determining the error of the fitted isodesmic $K_D$ for R221C.

**Figure supplement 9.** Averaging the conformational weights from different SAXS experiments.

modelling the conformational heterogeneity for each oligomer. The improvement in agreement with the SAXS data over the starting structures, also for individual conformers, suggests that the MD simulations contribute, not only by modelling the conformational heterogeneity, but also by simply relaxing the structure to a more accurate state.

The results described above show that the SAXS data fit well to an isodesmic model with a $K_D$ value close to that determined from CG-MALS. We wished to validate our approach further by comparing the SAXS-derived model of self-association with the CG-MALS data more directly. We therefore calculated the average molecular weight given by the isodesmic model with the $K_D$ of 1.3 μM that we obtained by fitting to the SAXS data and compared the results with the CG-MALS data (*Figure 3—figure supplement 4*). This analysis confirmed that the model of self-association derived from our analysis of the SAXS data is fully consistent with the independently measured CG-MALS data.

Having generated a conformational ensemble of each SPOP oligomer in agreement with the SAXS data, we proceeded to analyze the structures. Reweighting resulted in an increase in the radius of gyration ($R_g$) for almost all oligomeric species, suggesting that slightly more expanded conformations than those sampled with our modified version of Martini are more consistent with the SAXS data (*Figure 4a–b* and *Figure 4—figure supplement 1*). This expansion can be attributed both to a slight increase in the end-to-end distance for most oligomers (*Figure 4c–d* and *Figure 4—figure supplement 2*), as well as a slight increase in the average distance between the MATH and BTB/BACK domains for all oligomers upon reweighting (*Figure 4e–f*).

In order to investigate the global flexibility and compaction of SPOP oligomers, we fitted a power law to the average end-to-end distance ($R_{E-E}$) as a function of the number of subunits in the oligomer ($N$), $R_{E-E} = R_0 N^\nu$, where $R_0$ is the subunit segment size and $\nu$ is a scaling exponent (*Figure 4c*). The fit gave $R_0 = \sim 3.1$ nm and $\nu = 0.99$, showing a linear growth of the end-to-end distance with the number of subunits. This result is consistent with no significant curvature or compaction of the oligomers and, along with the narrow distribution of end-to-end distances for each oligomer (*Figure 4—figure supplement 2*), suggests that the SAXS data is compatible with a distribution of straight and relatively rigid SPOP oligomers, at least on length scales up to the ~180 nm of the 60-mer. The helical structure of larger oligomers, with ~16 subunits per turn, is evident as small periodic deviations from the fit (*Figure 4c*).

The MATH and BTB domains are connected through a ~20 residue long linker region (*Figure 1b*). We hypothesized that this linker may be flexible, allowing for reconfiguration of the MATH domains with respect to the crystal structure (*Zhuang et al., 2009*). We calculated the distances between the center-of-mass (COM) of the MATH domain and the COM of the BTB/BACK domains in the ensembles for every subunit of every oligomer. The distribution of these MATH-BTB/BACK distances reveal two populations overlapping with the two crystal structure configurations (*Figures 4e, i , and 5c*), where the MATH domains are in close proximity to the BTB/BACK domains (*Zhuang et al., 2009*). However, there is also a third population in which the MATH domains are extended away from the BTB/BACK domains, suggesting that the MATH-BTB linker is flexible and allows for movement out of the configurations observed in the crystal structure. Reweighting slightly increased the population of this extended state (*Figure 4e–f*). This flexibility in the configuration of the MATH domains gives rise to a broad distribution of distances between the substrate binding sites in neighbouring MATH domains, which is also slightly increased upon reweighting for all oligomers (*Figure 4g–h*). Both the overall rigidity of the oligomers and the flexibility of the MATH domains are also evident from visual inspection of the conformational ensemble of the 60-mer (*Figure 4j*).

To examine further whether the SAXS data support the observed flexibility of the MATH-BTB linker and repositioning of the MATH domains, we generated new ensembles of SPOP oligomers following the same protocol as above, but this time restraining the MATH domains to the BTB-BACK domains based on the configuration in the crystal structure using the elastic network model implemented in Martini. We calculated SAXS data from the generated ensembles and again fitted the

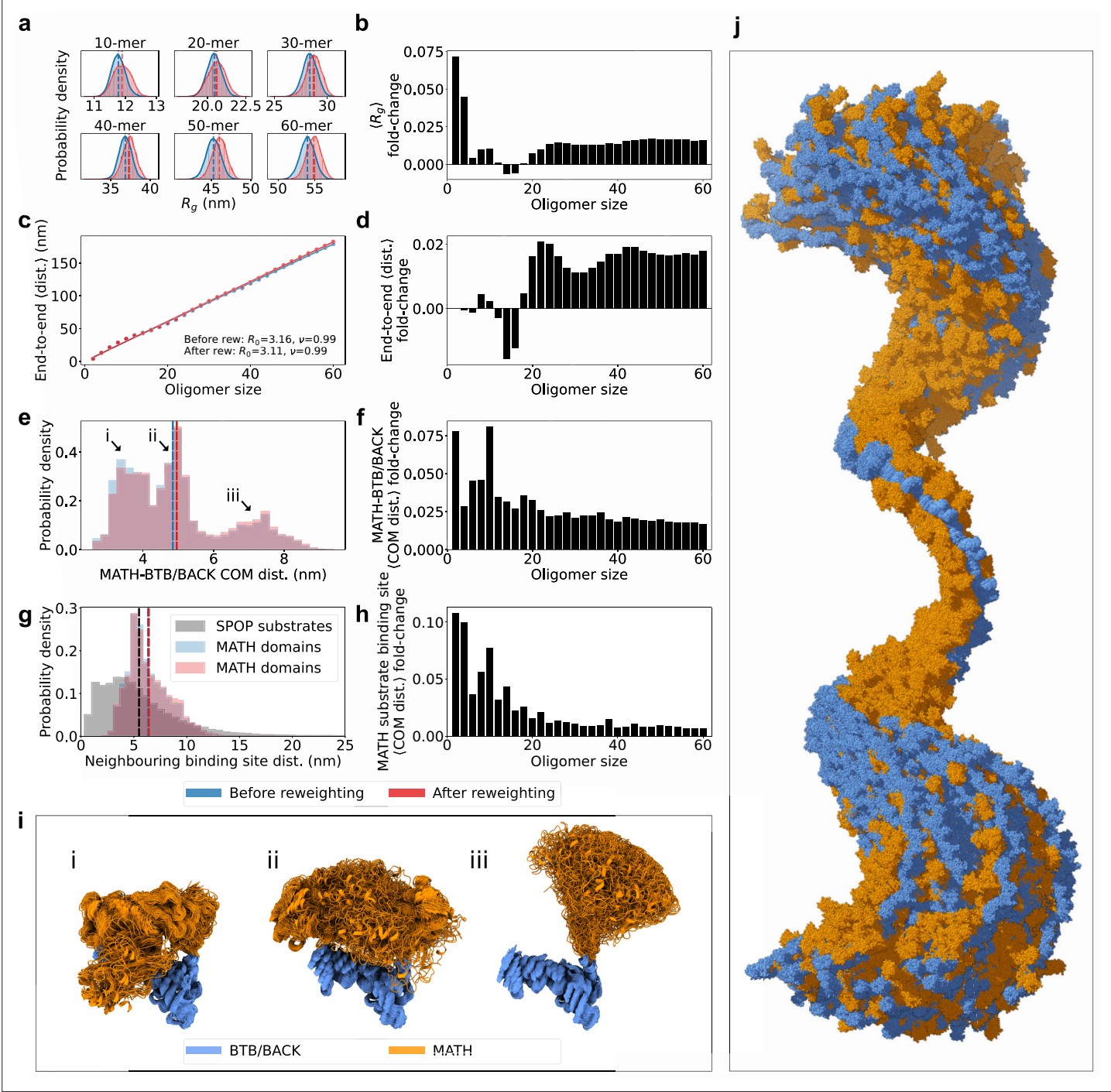

**Figure 4.** SPOP forms rigid, linear oligomers with flexible MATH domains in solution. (**a**) Probability distribution of the radius of gyration ($R_g$) calculated from ensembles of six representative SPOP oligomers before and after reweighting (see ***Figure 1*** for $R_g$ distributions for all oligomers). Dashed lines show the average values. (**b**) The fold-change in average $R_g$ after reweighting for all SPOP oligomers. (**c**) The average end-to-end distance calculated from ensembles of SPOP oligomers before and after reweighting (see ***Figure 2*** for distributions for all oligomers). Solid lines show the fit of a power law: $R_{E\text{-}E} = R_0 N^{\nu}$, where $R_{E\text{-}E}$ is the average end-to-end distance, $R_0$ is the subunit segment size, $N$ is the number of subunits in the oligomer, and $\nu$ is a scaling exponent. The fit gave $R_0$=3.16 nm, $\nu$=0.99 before reweighting and $R_0$=3.11 nm, $\nu$=0.99 after reweighting. (**d**) The fold-change in average end-to-end distance after reweighting for all SPOP oligomers. (**e**) Normalized histogram of distances between the center-of-mass (COM) of the MATH domain and the COM of the BTB/BACK domains in the same subunit before and after reweighting. The histogram contains the distances from every conformation of every subunit in every oligomer. (**f**) The fold-change in average MATH-BTB/BACK COM distance after reweighting for all SPOP oligomers. (**g**) Normalized histogram of COM distances between MATH substrate binding sites in neighbouring subunits (blue and red). The histogram

*Figure 4 continued on next page*

*Figure 4 continued*

contains the distances from every conformation of every subunit in every oligomer. In black, distances between neighbouring SPOP binding sites in seven SPOP substrate IDRs calculated from CALVADOS simulations. (**h**) The fold-change in average COM distance between neighbouring MATH substrate binding sites after reweighting for all SPOP oligomers. (**i**) Overlay of conformational ensembles corresponding to the three populations in panel (**e**) The structures are from all non-terminal subunits of the SPOP dodecamer and are superposed on the BTB/BACK domains. (**j**) Overlay of 151 randomly selected frames from the conformational ensemble of the SPOP 60-mer with atoms represented as spheres. Structures were superposed to the BTB/BACK domains in the four middle subunits. MATH domains are shown in orange and BTB/BACK domains are shown in blue.

The online version of this article includes the following figure supplement(s) for figure 4:

**Figure supplement 1.** $R_g$ distributions before and after reweighting.

**Figure supplement 2.** End-to-end distance distributions before and after reweighting.

**Figure supplement 3.** SPOP substrate motif-motif distances and motif-motif spacing.

---

isodesmic $K_D$ globally to the SAXS data, resulting in $K_D$=0.2±0.2 μM. The agreement with the SAXS data was substantially worse than for the original ensembles with the MATH domains unrestrained ($\chi^2_{r,\text{global}}$=4.38 and $\chi^2_{r,\text{global}}$=1.24 respectively), and the systematic deviations from the experimental SAXS profiles were clearly exacerbated (*Figure 5*). These results suggest that, first, the resolution of the SAXS data is high enough to distinguish between different configurations of the MATH domains and, second, that the SAXS data are indeed in better agreement with a model where the MATH-BTB linker is flexible. Taken together, our results support a model where, in solution, SPOP oligomers behave as rigid, helical structures with flexible MATH domains that can extend away from the BTB/BACK domains.

The comparison between the conformational ensembles generated with the MATH domains free or restrained also suggests that accurate conformational ensembles are necessary for accurate determination of the isodesmic $K_D$. Fitting the SAXS data using ensembles with the MATH domains restrained resulted in a lower isodesmic $K_D$, and calculating the agreement with SAXS for a range of isodesmic $K_D$ values revealed that there was no clear minimum in the $\chi^2_{r,\text{global}}$ for $K_D$ values > 0, which is also reflected in the large error range for the fitted $K_D$ (*Figure 5—figure supplement 1*). In line with this observation, validation with SAXS data at 15 μM protein revealed that improving the accuracy of the ensembles by reweighting also improved the accuracy of the fitted isodesmic $K_D$ independently of the fitted ensemble weights (*Figure 3—figure supplement 1d*).

To explore further how changes in the conformational ensemble would affect the agreement with the SAXS data, we used subsampling to generate ensembles with specific properties based on the ensembles with unrestrained MATH domains. To keep the comparison of ensembles unbiased by our previous fitting to the SAXS data, we used the isodesmic $K_D$=1.6 μM from CG-MALS for all comparisons with SAXS. First, we selected frames with lower average MATH-BTB/BACK COM distances (*Figure 5—figure supplement 3*). In line with the results from simulations with restrained MATH domains, this worsened the agreement with the SAXS data. In contrast, selecting frames with higher average MATH-BTB/BACK COM distance slightly improved the agreement with the SAXS data, in line with the results from reweighting (*Figure 5—figure supplement 4*). We also wished to test how sensitive the agreement with the SAXS data was to the overall shape of the oligomers. However, the conformational space that we could explore by subsampling the ensembles was limited by the rigidity of the oligomers. Despite this limitation, we subsampled ensembles with slightly higher and lower end-to-end distances than the original ensembles, corresponding to oligomers that are more or less extended than the original ensembles. Again consistent with the results from reweighting, ensembles with lower end-to-end distance resulted in slightly worse agreement with the SAXS data (*Figure 5—figure supplement 5*), while ensembles with higher end-to-end distance did not substantially change the agreement with the SAXS data (*Figure 5—figure supplement 6*). This result suggests that the SAXS data is less consistent with more compact SPOP oligomers, at least within the local part of conformational space explored here.

Self-association allows SPOP to bind disordered substrates that contain multiple SPOP binding motifs through multivalent interactions (*Pierce et al., 2016*). We hypothesized that the spacing between MATH domains in SPOP oligomers could be related to the spacing between SPOP binding motifs in disordered substrates. To investigate this, we selected five SPOP substrates with multiple SPOP binding motifs located in IDRs (SETD2 *Zhu et al., 2017*, SCAF1 *Theurillat et al., 2014*, SRC3 *Li et al., 2011*; *Geng et al., 2013*; *Janouskova et al., 2017*, Gli2, and Gli3 *Zhang et al., 2006*; *Zhang*

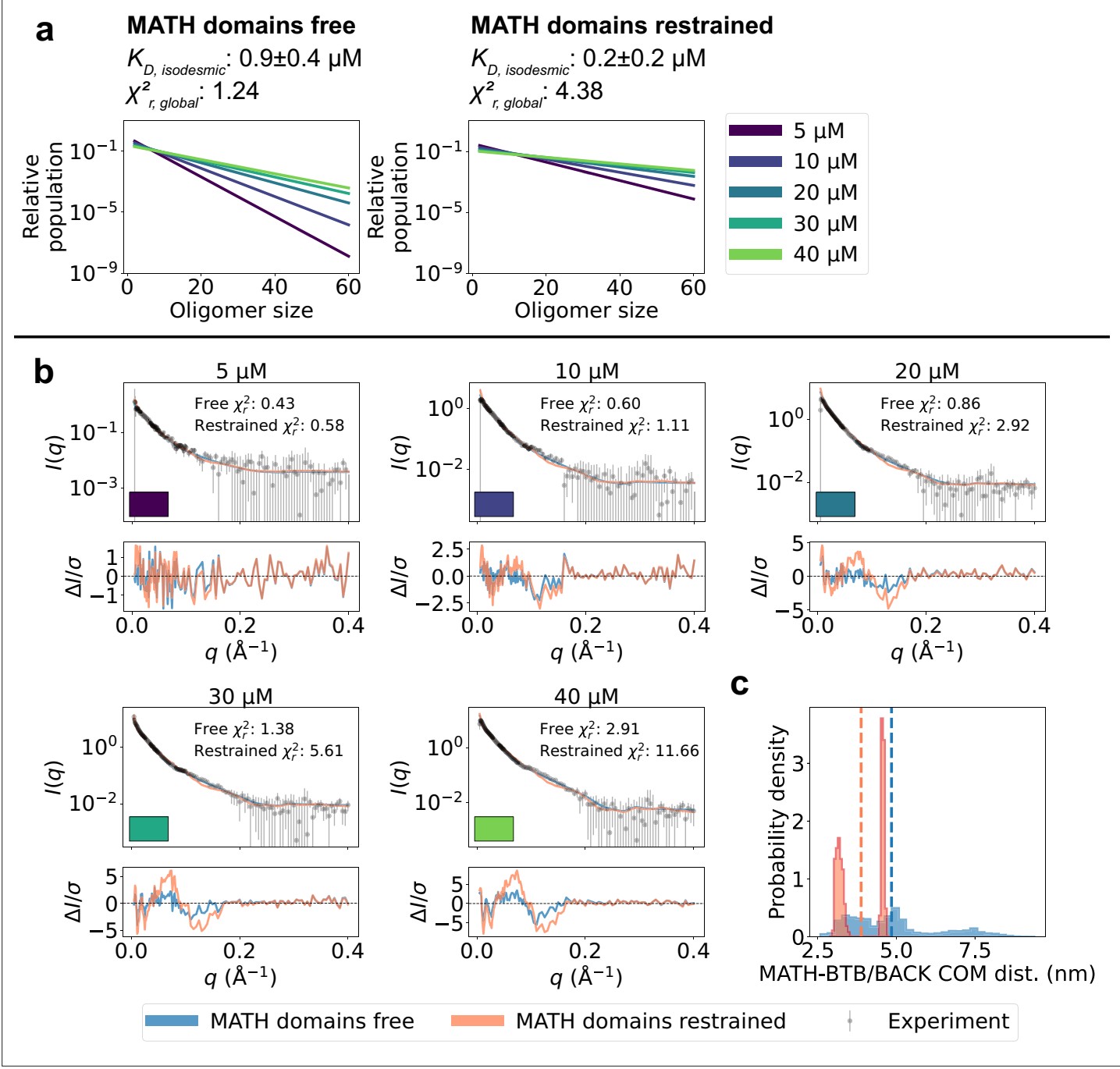

**Figure 5.** Unrestrained MATH domains give better agreement with SAXS data. Comparison of conformational ensembles with MATH domains either unrestrained (blue) or restrained to BTB/BACK domains based on the configuration in the crystal structure (orange). (**a**) Relative populations of oligomers for the protein concentrations used in SAXS experiments. Note the logarithmic scale. Populations are given by the isodesmic model with the $K_D$ noted above the plot. $K_D$ was fitted globally to the SAXS data in panel (**b**). $\chi^2_{r,\text{global}}$ quantifies the agreement with SAXS data in panel b for the two setups. (**b**) Agreement between experimental SAXS data and averaged SAXS data calculated from conformational ensembles of SPOP oligomers generated with the two setups. Oligomer populations are given by the isodesmic model (as shown in panel a). Error-normalized residuals are shown below the SAXS profiles and $\chi^2_r$ to each SAXS profile is shown on the plot. (**c**) Histogram of center-of-mass distances between MATH and BTB/BACK domains in the same subunit calculated from all conformations of all subunits of all oligomers. Average values are shown as dashed lines.

The online version of this article includes the following figure supplement(s) for figure 5:

**Figure supplement 1.** Determining the error of the fitted isodesmic $K_D$ with ensembles with MATH restrained.

**Figure supplement 2.** $R_g$ distributions from simulations with MATH free and MATH restrained.

*Figure 5 continued on next page*

**eLife** Research article

Structural Biology and Molecular Biophysics

**Figure supplement 3.** Subsampling compact MATH domains.

**Figure supplement 4.** Subsampling extended MATH domains.

**Figure supplement 5.** Subsampling low end-to-end distance.

**Figure supplement 6.** Subsampling high end-to-end distance.

*et al., 2009*) and ran coarse-grained simulations of their IDRs (seven IDRs in total) using CALVADOS, a one-bead-per-residue implicit solvent model that has been optimized to reproduce accurate global dimensions and transient interactions in IDPs (*Tesei et al., 2021*). We calculated the distances between neighbouring SPOP binding motifs in the simulations, and compared these with the distances between substrate-binding sites in neighbouring MATH domains given by our ensembles of SPOP oligomers (*Figure 4g*). This revealed substantial overlap between the two distributions, with a similar average distance between neighbouring binding sites in SPOP and in substrates, suggesting that the spacing of SPOP binding motifs in substrates may be evolutionarily optimized for multivalent binding to MATH domains.

Having analyzed the conformational properties of wild type SPOP and shown that the SAXS data are sensitive to the degree of self-association, we next wished to test whether our approach could capture the effects of mutations on SPOP self-association. We collected a concentration series of SAXS data on the SPOP mutant R221C, which has been identified in melanoma (*Krauthammer et al., 2012*) and colorectal cancer (*Giannakis et al., 2016*). R221C is located in the BTB-BTB interface, so we hypothesized that it may affect SPOP's propensity to self-associate. We used the same approach as for wild type to fit the isodesmic $K_D$ globally to the SAXS data, but without reweighting the conformational ensembles. For R221C, the isodesmic $K_D$ was fitted to 8.2±2.3 μM, which resulted in a reasonable fit to the SAXS data with $\chi^2_{global}$=1.79 (*Figure 3—figure supplement 7*), suggesting that the mutation results in a decreased propensity to self-associate compared with wild type ($K_D$=0.9 μM using a comparable approach or 1.3 μM when also reweighting the ensemble). Because R221C is located at the BTB-BTB interface, the 6–9 fold increase of the isodesmic $K_D$ (which relates to BACK-BACK dimerization) is perhaps surprising. While a long-range effect of R221C cannot be ruled out, an alternative mechanism may involve shifting the equilibrium of the BTB-BTB dimer, thus effectively decreasing the concentration of dimeric species available for self-association.

## Discussion

The ability of SPOP, a cancer-associated substrate adaptor in the ubiquitination machinery, to self-associate is important for its role in biology and disease. Characterizing the conformational ensemble of flexible and self-associating proteins such as SPOP from ensemble-averaged experiments is, however, difficult due to conformational and compositional heterogeneity. In one approach, SAXS data of mixtures may be attempted to be decomposed into contributions of individual components that may then be analysed separately (*Herranz-Trillo et al., 2017*; *Meisburger et al., 2021*). Here, we have developed an alternative 'forward modelling' approach to characterize proteins that undergo polydisperse oligomerization by self-consistently and globally fitting the distribution of oligomeric species and reweighting the conformational ensembles of the oligomers against SAXS data. A similar idea has recently been applied to study the self-association of tubulin using static structures as input (*Shemesh et al., 2021*). We recorded a concentration series of SAXS data on SPOP, which is known to form linear higher-order oligomers, and combined MD simulations with our approach to simultaneously refine conformational ensembles of thirty oligomeric states of SPOP along with the relative populations.

Our results suggest that SPOP oligomers are rigid, helical structures in solution and that the MATH-BTB linker is flexible, allowing for the extension of MATH domains away from the oligomer core. This is consistent with SPOP's proposed role in phase separation, as reconfiguration of the MATH domains could facilitate binding of substrates across multiple MATH domains and between different SPOP oligomers (*Pierce et al., 2016*; *Bouchard et al., 2018*). Indeed, the spacing of the MATH domains in our model of SPOP oligomers is consistent with the distances between motifs in ensembles of disordered SPOP substrates, based on coarse-grained simulations of disordered SPOP substrates. It has been suggested previously

that rigidity could play an important role in the phase separation of SPOP oligomers by ensuring a low conformational entropy penalty upon stacking linear oligomers with cross-bound substrates in the dense phase (*Schmit et al., 2020*). This is also consistent with the rigid structural model of SPOP oligomers proposed here. Our results also provide orthogonal evidence that SPOP self-association is described well by the isodesmic model, and that the isodesmic $K_D$ for BACK-BACK mediated self-association is in the low micromolar range, in agreement with previous measurements by CG-MALS (*Marzahn et al., 2016*). We also collected SAXS data and fitted the isodesmic $K_D$ for the SPOP mutant R221C. Our results suggest that SPOP R221C has a six- to ninefold decreased propensity to self-associate.

While the analysis of the SAXS data presented here does not strictly exclude the possibility that SPOP forms branched or otherwise non-linear oligomers, our results show that linear oligomers based only on the self-association interfaces known from existing crystal structures are consistent with SAXS data. Thus, linear oligomers seem to be the most plausible model based on this and other existing experimental evidence, for example that removal or mutation of either the BACK-BACK or BTB-BTB interface results in abolishment of higher-order self-association and that higher-order oligomers are formed through the self-association of SPOP dimers with every step of subunit-addition populated (*Marzahn et al., 2016*). Finally, as shown here, linear isodesmic self-association with the same $K_D$ provides a good fit to both SAXS and CG-MALS data (*Marzahn et al., 2016*).

The approach presented here to study SPOP can be extended to other polydisperse systems to characterize the distribution of oligomeric states and their conformational properties. However, there are a few limitations to be aware of; SAXS is a low-resolution technique, and may not be able to distinguish between all relevant conformations, a problem that is likely exacerbated here, as the contribution of many species to the SAXS signal may average out distinct features in the profile. One way to mitigate this problem is to construct multiple structural models, and test whether they show any difference in the agreement with the SAXS data. In the case of SPOP we used this approach to examine the flexibility of the MATH domain in SPOP[28–359].

Another limitation of the approach is the correlation between the fitted distribution of oligomeric states and the conformational properties of the oligomers. Here, we observed that a low isodesmic $K_D$ with large uncertainty was fitted when using more compact structures (MATH domains restrained), which suggests that the model can compensate for the underestimated dimensions of the proteins by increasing the populations of larger oligomers. Therefore, it is important to use prior conformational ensembles that are as accurate as possible. Additionally, it is important to include all the oligomeric species that make a substantial contribution to the SAXS data in the modelling. In the future, it might be relevant to include independent data reporting on the distribution of oligomeric species, such as from CG-MALS, when fitting SAXS data.

In the case of SPOP, we described the distribution of oligomers using the isodesmic self-association model, but this can be replaced by any model that describes the populations of the species in solution — with the caveat that there should not be too many free parameters to fit to the SAXS data. Similarly, the approach to generate prior conformational ensembles is not limited to MD simulations, and can be varied based on the system at hand. This flexibility in the modelling approach will make it useful to study other polydisperse systems in the future.

## Methods
### Protein expression and purification

The SPOP gene encoding residues 28–359 (His-SUMO-SPOP[28–359]) was expressed and purified as previously described (*Bouchard et al., 2018*). Briefly, His-SUMO-SPOP[28–359] was transformed into BL21-RIPL cells and expressed in auto-induction media (*Studier, 2005*). Cells were harvested, lysed, and cell debris was pelleted by centrifugation. The clarified supernatant was applied to a gravity Ni Sepharose resin equilibrated in resuspension buffer (30 mM imidazole, 1 M NaCl, pH 7.8). After washing with wash buffer (75 mM imidazole, 1 M NaCl, pH 7.8), the protein was eluted with a buffer containing 300 mM imidazole, 1 M NaCl, pH 7.8. One milligram of TEV protease was added to the eluted protein and the reaction was left to dialyze into 20 mM Tris pH 7.8, 300 mM NaCl, and 5 mM

DTT at 4 °C overnight. The cleaved protein was then further purified using a Superdex S200 size-exclusion chromatography column equilibrated with 20 mM Tris pH 7.8, 300 mM NaCl, and 5 mM DTT.

## Small-angle X-ray scattering

SAXS experiments were performed at the LIX-beamline (16-ID) of the National Synchrotron Light Source II (Upton, NY) (*DiFabio et al., 2016*). Data were collected at a wavelength of 1.0 Å, yielding an accessible scattering angle range of $0.006 < q < 3.2$ Å$^{-1}$, where $q$ is the momentum transfer, defined as $q = 4\pi \sin(\theta)/\lambda$, where $\lambda$ is the X-ray wavelength and 2θ is the scattering angle. Data with $q < 0.4$ Å$^{-1}$ were used for all analyses. Prior to data collection, SPOP was dialyzed into 20 mM Tris pH 7.8, 150 mM NaCl, and 5 mM DTT. Samples were loaded into a 1 mm capillary for ten 1 s X-ray exposures. Data were reduced at the beamline using the Python package *py4xs*.

## Molecular dynamics simulations with Martini

We ran coarse grained molecular dynamics simulations of six SPOP[28–359] oligomers ranging from the dimer to dodecamer (in steps of dimeric protomer subunits) using a beta version (3.0.4.17) of the Martini 3 force field (https://github.com/KULL-Centre/papers/tree/main/2020/TIA1-SAS-Larsen-et-al/Martini; *Souza et al., 2021*) and Gromacs 2020 (*Abraham et al., 2015*). We built the SPOP monomer structure using Modeller (*Sali and Blundell, 1993*) based on the crystal structure of the MATH and BTB domains (PDB: 3HQI) (*Zhuang et al., 2009*) and a crystal structure of the BACK domain (PDB: 4HS2) (*van Geersdaele et al., 2013*). We built the dimer structure by superposing two monomer structures to the crystal structure of the BTB-BTB dimer interface in 3HQI. We then built larger oligomers by iteratively adding dimer structures to the linear oligomer. Dimers were added by superposing the terminal BACK domain of the oligomer and a terminal BACK domain of the dimer to the structure of the BACK-BACK dimer (4HS2).

The starting structures were coarse grained using the Martinize2 python script. Elastic network restraints of 500 kJ mol$^{-1}$ nm$^{-2}$ between backbone beads within a 1.2 nm cut-off were applied with Martinize2 to keep folded domains intact and to hold oligomer subunits together. In the 'MATH free' model, we removed all elastic network restraints between MATH and BTB/BACK domains, between MATH and MATH domains, and in the linker region between MATH and BTB/BACK domains, while in the 'MATH restrained' model, we only removed elastic network restraints between MATH and MATH domains and in the linker region between MATH and BTB/BACK domains, but kept restraints between MATH and BTB/BACK domains. We added dihedral and angle potentials between side chains and backbone beads with the *-scfix* flag in Martinize2. Using Gromacs *editconf*, we placed the dimer and tetramer in a dodecahedral box. To keep the box volume small, larger oligomers were aligned with the principal axis of the system and placed in triclinic boxes that were thus elongated along the x-axis. To keep these oligomers from rotating and self-associating across the periodic boundary, we added soft harmonic position restraints of 5 J mol$^{-1}$ nm$^{-2}$ along the y- and z-axis to the backbone beads of the terminal BTB/BACK domains. We solvated the systems using the Insane python script (*Wassenaar et al., 2015*) and added 150 mM NaCl along with Na$^+$ ions to neutralize the systems. In the 'MATH free' system, we rescaled the $\epsilon$ of the Lennard-Jones potentials between all protein and water beads by a factor 1.06 to favour extension of the MATH domains into solution (*Thomasen et al., 2022*), while the unmodified Martini 3 beta v.3.0.4.17 was used for the 'MATH restrained' model.

Energy minimization was performed using steepest descent for 10,000 steps with a 30 fs time-step. Simulations were run in the NPT ensemble at 300 K and 1 bar using the Velocity-Rescaling thermostat (*Bussi et al., 2007*) and Parinello-Rahman barostat (*Parrinello and Rahman, 1981*). Non-bonded interactions were treated with the Verlet cut-off scheme. The cut-off for Van der Waals interactions and Coulomb interactions was set to 1.1 nm. A dielectric constant of 15 was used. We equilibrated the systems for 10 ns with a 2 fs time-step and ran production simulations for 60 μs with a 20 fs time-step, saving a frame every 1 ns.

After running the simulations, molecule breaks over the periodic boundaries were treated with Gromacs *trjconv* using the flags *-pbc mol -center*. Simulations were backmapped to all-atom using a modified version of the Backward algorithm (*Wassenaar et al., 2014*), in which simulation runs are excluded and energy minimization is shortened to 200 steps (*Larsen et al., 2020*). Every fourth simulation frame was backmapped for a total of 15,000 conformers in each backmapped ensemble.

## Constructing ensembles of larger SPOP oligomers

We constructed conformational ensembles of larger SPOP[28–359] oligomers with up to 60 subunits by joining together conformers from the all-atom backmapped ensembles of the SPOP dodecamer. Using ensembles of two input SPOP oligomers (SPOP 1 and SPOP 2) we started by removing the last subunit of SPOP 1 and the first subunit of SPOP 2 to ensure that the newly joined subunits were internal and not terminal. We then removed additional subunits from SPOP 2 to reach the desired length of the output oligomer. Then, we selected a random frame from SPOP 1 and SPOP 2, superposed the BTB/BACK domains of the last two subunits of SPOP 1 to the BTB/BACK domains of the first two subunits of SPOP 2, and deleted the first two subunits of SPOP 2. Next, we checked for clashes between the newly joined subunits (shortest interatomic distance <0.4 Å), and rejected the new frame if there was a clash. This approach ensured that the terminal subunits in the constructed oligomer were also the terminal subunits in the MD simulation of the dodecamer, while all internal subunits in the constructed oligomer were also internal in the MD simulation. This approach was repeated to create 15,000 structures of each larger oligomer.

## Calculating SAXS intensities from conformational ensembles

We calculated SAXS intensities from each of the 15,000 conformers in each of our all-atom ensembles of SPOP oligomers using Pepsi-SAXS (*Grudinin et al., 2017*). To avoid overfitting to the experimental SAXS data, we used fixed values for the parameters that describe the contrast of the hydration layer, $\delta\rho$=3.34 e/nm$^3$, and the volume of displaced solvent, $r_0/r_m$ = 1.025, that have been shown to work well for intrinsically disordered and multidomain proteins (*Pesce and Lindorff-Larsen, 2021*). The forward scattering ($I(0)$) was set equal to the number of subunits in the oligomer, in order to scale the SAXS intensities proportionally to the particle volume.

## The isodesmic self-association model and averaging of SAXS intensities

The experimental SAXS profiles of SPOP report on the average of a polydisperse mixture of oligomeric species in solution. The concentration of each oligomer should follow the isodesmic model where the concentration of the smallest subunit, the BTB-BTB dimer, is given by:

$$c_1 = \frac{2c_{tot}K_A + 1 - \sqrt{4c_{tot}K_A + 1}}{2c_{tot}K_A^2} \tag{1}$$

The concentration $c_i$ of any larger oligomer with $i$ subunits can be calculated given $c_1$ and the concentration of oligomer $i$–1, $c_{i-1}$:

$$c_i = K_A c_{i-1} c_1 \tag{2}$$

$K_A$ is the isodesmic association constant and $c_{tot}$ is the total concentration of protomers. Here we assume that the SPOP BTB-BTB dimer is always fully formed (*Marzahn et al., 2016*) and $c_{tot}$ in *Equation 1* is thus half of the total protein concentration reported for the SAXS experiments, which refers to the SPOP monomer concentration. Given the concentration $c_i$ of each oligomer $i$ from the isodesmic model, we can calculate the volume fraction $\phi_i$ of the oligomer:

$$\phi_i = \frac{ic_i}{\sum_i^N ic_i} \tag{3}$$

The average SAXS intensities from the mixture of oligomers $\langle I \rangle_{\mathrm{mix}}$ are then given by:

$$\langle I \rangle_{\mathrm{mix}} = \sum_i^N \langle I \rangle_{i,\mathrm{ensemble}} \phi_i \tag{4}$$

where $\langle I \rangle_{i,\mathrm{ensemble}}$ is the conformationally averaged SAXS intensity of oligomer $i$. Note that the magnitude of the SAXS intensities calculated with Pepsi-SAXS were set to be proportional to the number of subunits in the oligomer, so given *Equations 3 and 4* the total contribution of each oligomer to the averaged SAXS intensity is proportional to $i^2 c_i$.

## Self-consistent optimization of isodesmic model parameters and conformational ensemble weights

The algorithm we developed to self-consistently optimize the isodesmic distribution of oligomer concentrations and reweight the conformational ensemble of each oligomer against SAXS data consists of three iterative steps: (1) fitting the scale and constant background of the SAXS data, (2) fitting the isodesmic $K_A$, and (3) reweighting the conformational ensemble of each oligomer using BME reweighting. We used a concentration series of SAXS experiments, to which the isodesmic $K_A$ was fitted globally, and only subsequently transformed the $K_A$ to the $K_D$ ($K_D = 1/K_A$) for reporting our results.

### Step 1: Fitting the SAXS scale and constant background

The following step was repeated for each SAXS experiment in the concentration series. The concentration of each oligomer was calculated using the isodesmic model with the given $c_{tot}$ (**Equations 1 and 2**). The average SAXS intensities $\langle I \rangle_{mix}$ from all oligomers were then calculated using **Equations 3 and 4**. The scale and constant background ($cst$) of $\langle I \rangle_{mix}$ were fitted to the experimental SAXS intensities, $I_{exp}$, using least-squares linear regression weighted by the experimental errors (*LinearRegression* function in scikit-learn **Pedregosa et al., 2011**):

$$I_{exp} = scale\langle I \rangle_{mix} + cst \tag{5}$$

In practice, to avoid modifying the SAXS scale and constant background for every conformer in our ensembles, we instead performed the inverse operation on the experimental SAXS profile:

$$I_{exp,fit} = \frac{I_{exp} - cst}{scale} \tag{6}$$

and propagated the experimental errors $\sigma_{exp}$ accordingly:

$$\sigma_{exp,fit} = \frac{\sigma_{exp}}{|scale|} \tag{7}$$

### Step 2: Fitting the isodesmic model

The isodesmic $K_A$ was fitted globally to the concentration series of SAXS experiments using Metropolis Monte Carlo (**Metropolis et al., 1953**) with simulated annealing. For each Monte Carlo step, we generated a new random $K_A$ with a Gaussian probability distribution centered around the previous $K_A$, calculated new oligomer concentrations and corresponding $\langle I \rangle_{mix}$ for each SAXS experiment in the concentration series using **Equations 1–4**, and for each SAXS experiment calculated the reduced $\chi^2$, $\chi_r^2$, as:

$$\chi_r^2 = \frac{1}{m} \sum_j^m \frac{(\langle I \rangle_{j,mix} - I_{j,exp})^2}{\sigma_{j,exp}^2} \tag{8}$$

where $m$ is the number of SAXS intensities $j$ in the SAXS profile. We then calculated the average of the $\chi_r^2$-values across the SAXS concentration series to get the global $\chi_r^2$, $\chi_{r,global}^2$, as the number of intensities was the same in each SAXS profile. Next, we evaluated the acceptance criterion by calculating:

$$\alpha = exp\left(-\frac{\chi_{new,r,global}^2 - \chi_{old,r,global}^2}{T}\right) \tag{9}$$

where $\chi_{new,r,global}^2$ and $\chi_{old,r,global}^2$ are the from the current and previous Monte Carlo step respectively and $T$ is the simulated annealing temperature. If $\alpha > 1$, we accepted the new $K_A$. If $\alpha=1$, we generated a random number, *rand*, between 0 and 1, and if $\alpha > rand$, accepted the new $K_A$. Otherwise, we kept the $K_A$ from the previous Monte Carlo step. Finally, we decreased $T$ for the next Monte Carlo step.

## Step 3: Reweighting the conformational ensemble

The following step was repeated for each SAXS experiment in the concentration series. We calculated the oligomer concentrations using the isodesmic model given the new $K_A$ determined in step 2. For each oligomer $i$, we extracted a SAXS profile for BME reweighting from the experimental profile using the following method: we calculated the average SAXS profile from the ensembles as in **Equation 4** but leaving out oligomer $i$ from the sum to get $\langle I \rangle_{\mathrm{mix,rest}}$. Next, we determined the contribution of species $i$ to the experimental SAXS intensity as:

$$\langle I \rangle_{i,\mathrm{extr}} = \frac{I_{\mathrm{exp}} - \langle I \rangle_{\mathrm{mix,rest}}}{\phi_i} \tag{10}$$

where $I_{\mathrm{exp}}$ is the experimental SAXS intensity and $\phi_i$ is the volume fraction of oligomer $i$. We then propagated the error $\sigma_{i,\mathrm{extr}}$ on $\langle I \rangle_{i,\mathrm{extr}}$ from both the errors on the experimental SAXS intensities and the errors on the calculated SAXS intensities, which we determined using block error analysis (**Flyvbjerg and Petersen, 1989**). The propagated errors were given by:

$$\sigma_{i,\mathrm{extr}} = \frac{\sqrt{\sigma_{\mathrm{exp}}^2 + \sum_r^N (\sigma_{r,\mathrm{block}} \phi_r)^2}}{\phi_i} \tag{11}$$

where the sum $r$ to $N$ runs over all oligomers that contributed to $\langle I \rangle_{\mathrm{mix,rest}}$, $\sigma_{\mathrm{exp}}$ is the error on the experimental SAXS intensity, $\sigma_{r,\mathrm{block}}$ is the error on the average SAXS intensity calculated from the ensemble of oligomer $r$ prior to reweighting using block error analysis (https://github.com/fpesceKU/BLOCKING; **Pesce, 2023**), and $\phi_i$ is the volume fraction of oligomer $i$. The conformational ensemble of oligomer $i$ was then reweighted against this extracted SAXS profile using BME reweighting (**Bottaro et al., 2020**), in which a set of ensemble weights $w$ are obtained by minimizing the function:

$$\mathcal{L}(w_1...w_n) = \frac{m}{2} \chi_r^2(w_1...w_n) - \theta S_{\mathrm{rel}}(w_1...w_n) \tag{12}$$

where $n$ is the number of ensemble conformations, $m$ is the number of experimental observables (in this case the number of SAXS intensities in the profile), $\chi_r^2$ quantifies the agreement between $\langle I \rangle_{\mathrm{ensemble}}$ and $\langle I \rangle_{\mathrm{extr}}$, $S_{\mathrm{rel}}$ is the relative Shannon entropy that quantifies the deviation of the new weights from the initial weights, $w^0$, and $\theta$ is a scaling parameter that quantifies the confidence in the experimental data versus the prior ensemble. $\chi_r^2$ is given by:

$$\chi_r^2(w_1...w_n) = \frac{1}{m} \sum_j^m \frac{\sum_k^n (w_k I_{j,k,\mathrm{ensemble}} - \langle I \rangle_{j,\mathrm{extr}})^2}{\sigma_{j,\mathrm{extr}}^2} \tag{13}$$

where $I_{j,k,\mathrm{ensemble}}$ is the SAXS intensity $j$ calculated from the conformer $k$ of the ensemble. $S_{\mathrm{rel}}$ is given by:

$$S_{\mathrm{rel}} = -\sum_k^n w_k log\left(\frac{w_k}{w_k^0}\right) \tag{14}$$

Using the ensemble weights obtained from BME reweighting, we calculated new weighted average SAXS intensities, $\langle I \rangle_{i,\mathrm{ensemble}}$, from the ensemble of oligomer $i$. The process of extracting a SAXS profile followed by BME reweighting was repeated for each oligomer.

## Optimization parameters

The three steps described above were repeated iteratively to converge on self-consistent values of the SAXS scale and constant background, the isodesmic $K_A$, and the ensemble weights for each oligomer species. As the SAXS profile against which the ensemble of each oligomer was reweighted is a function of the ensemble weights of all other oligomeric species, we wished to reweight the ensembles only slightly in initial iterations, and then gradually increase the degree of reweighting as the conformational weights and isodesmic $K_A$ converged. We achieved this by starting with a high value of $\theta$ (**Equation 12**) and then gradually decreasing $\theta$ each iteration. The fraction of effective frames, $\phi_{\mathrm{eff}}$, given by $\exp(S_{\mathrm{rel}})$, provides a measure of the fraction of the initial ensemble that is retained after reweighting. At every iteration, we checked whether the ensemble of each oligomer had reached a

$\phi_{\text{eff}}$ below a set cut-off, after which $\theta$ was no longer decreased for that specific oligomer. Thus, the overall degree of reweighting could be tuned through selection of this $\phi_{\text{eff}}$-cut-off.

We ran the optimization scheme for 1000 iterations starting with $\theta$=100 and decreasing $\theta$ by 2% every iteration. The simulated annealing of the isodesmic $K_A$ was run from $T$=10 to $T$=0.1 every iteration, with $T$ decreased by 30% every Monte Carlo step, and with a standard deviation of 0.1 µM$^{-1}$ for the Gaussian probability distribution used to generate the new $K_A$. The step was repeated if $K_A \leq$ 0 was generated.

## Preventing overfitting of ensemble weights

We ran the optimization with a range of $\phi_{\text{eff}}$-cut-offs from 0.1 to 1. To prevent overfitting, we aimed to choose a value of $\phi_{\text{eff}}$ that retained as much of the prior ensemble as possible (high $\phi_{\text{eff}}$) while not sacrificing substantial improvement in the fit to the SAXS data (low $\chi^2_{r,\text{global}}$). As an additional approach to prevent overfitting, we left out the SAXS experiment recorded with 15 µM protein from the optimization, and used it as validation for the determined weights (averaged as explained in the next section) and isodesmic $K_A$ at different values of the $\phi_{\text{eff}}$-cut-off. For each $\phi_{\text{eff}}$-cut-off, we fitted only the SAXS scale and constant background to the 15 µM SAXS experiment. We tested the effect of using the fitted $K_A$ and ensemble weights in combination, but also the effect of using only the fitted $K_A$ or ensemble weights independently. Although in all cases the fitted $K_A$ and ensemble weights combined improved the fit to the SAXS data compared with the initial weights and $K_A$, $\phi_{\text{eff}}$-cut-off=0.4 was the lowest value of $\phi_{\text{eff}}$ where the fit was not improved by replacing the fitted weights with uniform weights in combination with the fitted $K_A$ (*Figure 3—figure supplement 1*). Thus, we selected the conformational weights and isodesmic $K_A$ determined with $\phi_{\text{eff}}$-cut-off=0.4 to avoid overfitting the ensemble weights.

## Averaging the conformational weights from different SAXS experiments

The optimization scheme outputs a set of conformational weights for each SAXS experiment in the concentration series. We combined these conformational weights to obtain a single set of weights for further analysis, under the assumption that the conformational properties of each SPOP oligomer are independent of protein concentration. The distribution of oligomeric species from the isodesmic model depends on the protein concentration. Thus, each SAXS experiment does not contain the same amount of information on every oligomer; SAXS experiments at lower concentrations have a relatively smaller contribution from large oligomers and vice versa. Therefore, we weighted the averaging of the conformational weights to reflect this mismatch in information. The average weight of conformation $k$ of oligomer $i$ was calculated as:

$$\langle w \rangle_{k,i} = \sum_{l}^{o} w_{k,i,l} \rho_{i,l} \tag{15}$$

where $w_{k,i,l}$ is the weight of conformer $k$ of oligomer $i$ from reweighting against SAXS experiment $l$ and $\rho_{i,l}$ is the contribution of oligomer $i$ to SAXS experiment $l$ relative to the contribution of oligomer $i$ to the other SAXS experiments in the concentration series, given by:

$$\rho_{i,l} = \frac{1}{\sum_{l}^{o} \frac{i^2 c_{i,l}}{\sum_{i}^{N} i^2 c_{i,l}}} \frac{i^2 c_{i,l}}{\sum_{i}^{N} i^2 c_{i,l}} \tag{16}$$

where $c_{i,l}$ is the concentration of oligomer $i$ in SAXS experiment $l$ given by the isodesmic model. For a plot of the contributions $\rho_{i,l}$, see *Figure 3—figure supplement 9*.

## Determining the error of the fitted isodesmic $K_D$

To determine the uncertainty of the isodesmic $K_D$ fitted with our optimization scheme, we scanned a range of $K_D$ values around the fitted $K_D$ and determined the $\chi^2_{r,\text{global}}$ to the concentration series of SAXS data. We used the same ensemble weights for every value of $K_D$, and only fitted the scale and constant background to the SAXS data. We then defined the error of the fitted $K_D$ to include all $K_D$ values that gave a $\chi^2_{r,\text{global}}$ to the SAXS data within 10% of the minimum $\chi^2_{r,\text{global}}$.

## Analysis of SPOP conformational ensembles

$R_g$ was calculated from ensembles using the *gyrate* function in Gromacs. End-to-end distances were calculated from ensembles as the distances between the center-of-mass (COM) of the BTB/BACK domains in the terminal subunits using the *compute_center_of_mass* function in MDTraj (**McGibbon et al., 2015**) and the *linalg.norm* function in NumPy (**Harris et al., 2020**). We fitted ensemble averaged end-to-end distances against oligomer size (number of subunits) with a power law: $R_{E\text{-}E} = R_0 N^\nu$, where $R_0$ is the subunit segment size, $N$ is the number of subunits in the oligomer, and $\nu$ is a scaling exponent, using the *curve_fit* function in SciPy (**Virtanen et al., 2020**). To subsample ensembles with extended or compacted oligomers, frames were selected with $R_{E\text{-}E} > \max(R_{E\text{-}E}) - \frac{\max(R_{E\text{-}E}) - \langle R_{E\text{-}E}\rangle}{2}$ or $R_{E\text{-}E} < \min(R_{E\text{-}E}) + \frac{\langle R_{E\text{-}E}\rangle - \min(R_{E\text{-}E})}{2}$ respectively, where $\max(R_{E\text{-}E})$ and $\min(R_{E\text{-}E})$ are the maximum and minimum over all frames of the ensemble and $\langle R_{E-E}\rangle$ is the ensemble average. MATH-BTB/BACK COM distance was calculated from ensembles as the distance between the COM of the MATH domain and BTB/BACK domains in every subunit using the *compute_center_of_mass* function in MDTraj and the *linalg.norm* function in NumPy. The histogram of MATH-BTB/BACK COM distances shows values for all conformations of all subunits of all oligomers. To subsample ensembles with compacted or extended MATH domains, frames were selected with an average MATH-BTB/BACK COM distance over all subunits <4.4 nm or >5.2 nm, respectively. The COM distance between substrate binding sites in neighbouring MATH domains was calculated from ensembles using the *distance* function in Gromacs. The MATH substrate binding site was defined as residue Arg70, Tyr87, Ser119, Tyr123, and Lys129-Phe133. The histogram of MATH binding site COM distances shows values for all conformations of all subunits of all oligomers. Structures for *Figure 4i* were selected by fitting three Gaussians to the histogram in *Figure 4e* (after reweighting) using SciPy *curve_fit* and for each Gaussian selecting conformers within 0.1σ of the mean. All visualizations of protein structures were made with ChimeraX (**Pettersen et al., 2021**). To examine the agreement of single frames drawn from the ensembles with SAXS data, we drew a random frame from the ensemble of each oligomer and scanned the isodesmic $K_D$ from 0.01 to 100 μM (with 10,000 log-spaced steps) to select the $K_D$ that gave the optimal agreement with the SAXS concentration series based on $\chi^2_{r,\text{global}}$. The SAXS scale and constant background were fitted for each $K_D$. This procedure was repeated for 10,000 iterations. The same procedure was performed with oligomer structures constructed prior to the MD simulations.

## Dimer-oligomer equilibria and averaging of SAXS intensities

For dimer-oligomer equilibria, the total concentration of BTB-mediated dimer subunits (both free and in oligomers), $c_{\text{tot,dimer}}$, was assumed to be half of the total SPOP monomer concentration. We determined the equilibrium dimer concentration, $c_{\text{dimer}}$, and oligomer concentration, $c_i$, for a given association constant $K_A$ using:

$$K_A = \frac{c_i}{c_{\text{dimer}}^{(i/2)}} \tag{17}$$

and the equation for conservation of mass:

$$c_i = \frac{c_{\text{tot,dimer}} - c_{\text{dimer}}}{i/2} \tag{18}$$

where, $i$ is the number of subunits in the oligomer. The averaged SAXS intensities $\langle I\rangle_{\text{mix}}$ were then calculated as:

$$\langle I\rangle_{\text{mix}} = \phi_{\text{dimer}}\langle I\rangle_{\text{dimer,ensemble}} + \phi_i\langle I\rangle_{i,\text{ensemble}} \tag{19}$$

where $\langle I\rangle_{\text{dimer,ensemble}}$ and $\langle I\rangle_{i,\text{ensemble}}$ are the ensemble averaged SAXS intensity for the dimer and oligomer, and $\phi_{\text{dimer}}$ and $\phi_i$ are the volume fractions of the dimer and oligomer calculated based on the concentrations and number of subunits. For each possible dimer-oligomer equilibrium, we scanned $K_A$ values from $10^{-12}$–$10^{12}$ μM$^{-1}$ and selected the $K_A$ that gave the optimal agreement with the SAXS concentration series based on $\chi^2_{r,\text{global}}$. The SAXS scale and constant background were fitted for each $K_A$.

## Molecular dynamics simulations with CALVADOS

We selected five SPOP substrates with at least 8 SPOP binding motifs (*Cuneo and Mittag, 2019*) for simulations (SETD2 *Zhu et al., 2017*, SCAF1 *Theurillat et al., 2014*, SRC3 *Li et al., 2011*; *Geng et al., 2013*; *Janouskova et al., 2017*, Gli2, and Gli3 *Zhang et al., 2006*; *Zhang et al., 2009*). We selected the IDRs of these proteins based on low Alphafold pLDDT scores and pairwise alignment errors (*Jumper et al., 2021*). We ran coarse-grained simulations of these with CALVADOS 2 (*Tesei et al., 2021*; *Tesei and Lindorff-Larsen, 2023*). Simulations were run at 298 K, with an ionic strength of 150 mM, and pH 7.2 for determining the partial charge of histidine side-chains. Simulations were run for $3 \times 10^3 N_{res}^2$ steps, where $N_{res}$ is the number of residues, using a 10 fs time-step (*Tesei and Lindorff-Larsen, 2023*). Frames were saved every $3N_{res}^2$ steps to obtain weakly correlated frames. We used a 2 nm cutoff for the Ashbaugh-Hatch potential and a 4 nm cutoff for the Debye-Hückel potential. All simulations were started from a linear arrangement of the protein chain, except for simulations of the two longest IDRs, SCAF1 IDR and SETD2 IDR 1, which were started from an Archimedean spiral arrangement. Simulations were performed with HOOMD-blue 2.9.3 (*Anderson et al., 2020*).

## Analysis of motif spacing in SPOP substrates

We identified SPOP binding motifs in the substrate sequences as five consecutive positions with residues 1: GAVLIMWFPC, 2: STCYNQDEHR, 3: ST, 4: STCYNQDEHR, 5: ST or 1: GAVLIMWFPC, 2: STCYNQDEHR, 3: ST, 4: ST, 5: STCYNQDEHR, where each set of amino acids are allowed at the given position (*Zhuang et al., 2009*; *Cuneo and Mittag, 2019*). We calculated a histogram of all distances between neighbouring motifs in the SPOP susbstrate sequences over the CALVADOS simulations. Distances were calculated between the middle residue beads of the neighbouring motifs using the *compute_contacts* function in MDTraj. We also calculated the average distance, $R$, between each neighbouring motif and fit this with a power law $R = R_0 N^\nu$, where $R_0$ is the segment size, $N$ is the number of residues spacing the two motifs, and $\nu$ is a scaling exponent, using the *curve_fit* function in SciPy.

## Fitting CG-MALS data

Given the concentration of each oligomer from the isodesmic model, the average molecular weight, as measured by CG-MALS, was calculated as:

$$\langle MW \rangle = \sqrt{\frac{1}{N\sum_i^N c_i} \sum_i^N (iMW_{monomer})^2 c_i} \tag{20}$$

where $N$ is the number of oligomers, $c_i$ is the concentration of oligomer $i$ given by the isodesmic model and $MW_{monomer}$ is the molecular weight of the subunit of oligomerization.

We fitted the isodesmic $K_D$ and $MW_{monomer}$ to the CG-MALS data from *Marzahn et al., 2016*. The CG-MALS data consists of two merged data-sets, so we allowed a different $MW_{monomer}$ for each of the two merged data-sets to absorb uncertainties from determination of the protein concentrations. The $K_D$ was fitted globally to the two merged data-sets, and the error of the fit on the $K_D$ was set to two standard deviations. Fitting was done with the *curve_fit* function in SciPy.

## Acknowledgements

This work was supported by the Lundbeck Foundation BRAINSTRUC structural biology initiative (R155-2015-2666, to K.L.-L.), the PRISM (Protein Interactions and Stability in Medicine and Genomics) centre funded by the Novo Nordisk Foundation (NNF18OC0033950, to K.L.-L.), by NIH grant R01GM112846 (to T.M.) and by the American Lebanese Syrian Associated Charities (to T.M.). We acknowledge access to computational resources from the ROBUST Resource for Biomolecular Simulations (supported by the Novo Nordisk Foundation; NNF18OC0032608), the Danish National Supercomputer for Life Sciences (Computerome), and the Biocomputing Core Facility at the Department of Biology, University of Copenhagen. We thank Melissa R Marzahn and Erik W Martin for the generation of preliminary data. We thank Shirish Chodankar for assistance with SAXS data collection and reduction. The LiX beamline is part of the Center for BioMolecular Structure (CBMS), which is primarily supported by the National Institutes of Health, National Institute of General Medical Sciences (NIGMS) through a P30

Grant (P30GM133893), and by the DOE Office of Biological and Environmental Research (KP1605010). LiX also received additional support from NIH Grant S10 OD012331. As part of NSLS-II, a national user facility at Brookhaven National Laboratory, work performed at the CBMS is supported in part by the U.S. Department of Energy, Office of Science, Office of Basic Energy Sciences Program under contract number DE-SC0012704.

# Additional information

### Competing interests

Tanja Mittag: was a consultant for Faze Medicines, Inc. The other authors declare that no competing interests exist.

### Funding

| Funder | Grant reference number | Author |
| --- | --- | --- |
| Lundbeckfonden | R155-2015-2666 | Kresten Lindorff-Larsen |
| Novo Nordisk Fonden | NNF18OC0033950 | Kresten Lindorff-Larsen |
| National Institutes of Health | R01GM112846 | Tanja Mittag |
| American Lebanese Syrian Associated Charities | | Tanja Mittag |
| Novo Nordisk Fonden | NNF18OC0032608 | Kresten Lindorff-Larsen |
| National Institutes of Health | P30GM133893 | Tanja Mittag |
| DOE Office of Science's Biological and Environmental Research | KP1605010 | Tanja Mittag |
| National Institutes of Health | OD012331 | Tanja Mittag |

The funders had no role in study design, data collection and interpretation, or the decision to submit the work for publication.

### Author contributions

F Emil Thomasen, Conceptualization, Software, Formal analysis, Validation, Investigation, Visualization, Methodology, Writing - original draft, Writing – review and editing; Matthew J Cuneo, Investigation, Writing – review and editing; Tanja Mittag, Conceptualization, Resources, Supervision, Funding acquisition, Writing – review and editing; Kresten Lindorff-Larsen, Conceptualization, Resources, Supervision, Methodology, Project administration, Writing – review and editing

### Author ORCIDs

F Emil Thomasen http://orcid.org/0000-0002-2096-4873
Matthew J Cuneo http://orcid.org/0000-0002-1475-6656
Tanja Mittag http://orcid.org/0000-0002-1827-3811
Kresten Lindorff-Larsen http://orcid.org/0000-0002-4750-6039

### Decision letter and Author response

Decision letter https://doi.org/10.7554/eLife.84147.sa1
Author response https://doi.org/10.7554/eLife.84147.sa2

# Additional files

### Supplementary files

• MDAR checklist

## Data availability

Code and data is available at https://github.com/KULL-Centre/_2022_Thomasen_SPOP (copy archived at swh:1:rev:be995dd615079fe8b4fbb86941160d519429ee4c). Simulation data is available at https://doi.org/10.17894/ucph.ef999f72-b5e8-45c4-835f-3e49619a0f91. Plasmids are available from Addgene (plasmid IDs 194115 and 194116).

The following dataset was generated:

| Author(s) | Year | Dataset title | Dataset URL | Database and Identifier |
|---|---|---|---|---|
| Emil Thomasen F, Cuneo MJ, Mittag T, Lindorff-Larsen K | 2022 | Supporting data for Conformational and oligomeric states of SPOP from small-angle X-ray scattering and molecular dynamics simulations | https://doi.org/10.17894/ucph.ef999f72-b5e8-45c4-835f-3e49619a0f91 | Electronic Research Data Archive at University of Copenhagen, 10.17894/ucph.ef999f72-b5e8-45c4-835f-3e49619a0f91 |

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
