## [Editor Report]

In this important paper, the authors have developed an approach for simultaneously optimizing the conformational ensemble and degrees of oligomerization, and this has been tested by applying it to a specific protein (SPOP). Comparison of the quality of fits with different models also provides valuable insights into structural features important to the assembly of oligomers. The approach, presented with compelling experimental support, is potentially applicable to other systems as well.

---

## [Decision Letter]

**Decision letter after peer review:**

Thank you for submitting your article "Conformational and oligomeric states of SPOP from small-angle X-ray scattering and molecular dynamics simulations" for consideration by *eLife*. Your article has been reviewed by 3 peer reviewers, one of whom is a member of our Board of Reviewing Editors, and the evaluation has been overseen by Volker Dötsch as the Senior Editor. The following individual involved in the review of your submission has agreed to reveal their identity: Sichun Yang (Reviewer #2).

Essential revisions:

1) Further discuss the potential involvement of non-linear structures and functional implication of the structural features (e.g., impact on LLPS).

2) Further clarification of the fitting procedure, including, for example, the importance of sampling different conformations, consideration of violation of the isodesmic model, and alternative experimental validation of the conformation ensemble.

*Reviewer #1 (Recommendations for the authors):*

I think the approach and results are well organized and presented. I don't have much to add. The only suggestion I have is to add some discussions regarding the biological implications of the rigid structures established from this work. For example, to what degree such rigidity is relevant to the phase separation behaviors of SPOP?

*Reviewer #3 (Recommendations for the authors):*

The work could go further in more thoroughly testing each of the assumptions in light of the experimental data:

Does the SAXS data preclude alternate association modes that would give rise to a significant fraction of non-linear structures? The introduction mentions the role of SPOP in condensate formation and one could imagine shorter oligomers associating in different ways than simply growing into single long chains.

How would the fit to the SAXS data change if oligomer sizes are distributed differently, not following an isodesmic model?

How does the sampling of the conformational ensembles affect the fits? One question would be how the results depend on the number of snapshots extracted from the simulations. Another question would be how consistent the generated domain-domain configurations are with the experimental data. The latter question is addressed in part by showing that a truncated ensemble where the MATH and BTB/BACK domains are in close contact does not lead to a good agreement. What about ensembles that only consist of (i) and (iii), or ensembles that consist only of (ii) and (iii)?

Even though there is good agreement between the computational models and the SAXS data, further experimental validation of the proposed oligomers with different techniques may be warranted. While such experiments are likely beyond the scope of this work, some ideas about such experiments may be helpful.

---

## [Author Response]

Essential revisions:1) Further discuss the potential involvement of non-linear structures and functional implication of the structural features (e.g., impact on LLPS).

To address this point, we have added a Discussion section mentioning the possibility of non-linear structures to the manuscript. Additionally, we have added a short discussion of the functional implications of rigid oligomers, in the context of previous work on SPOP phase separation with disordered substrates (https://doi.org/10.1021/jacs.9b10066). We have also performed coarse-grained molecular dynamics simulations of several disordered SPOP substrates to investigate the relationship between the spacing of binding motifs in SPOP substrates and the substrate binding sites in SPOP oligomers given by our proposed conformational ensembles.

2) Further clarification of the fitting procedure, including, for example, the importance of sampling different conformations, consideration of violation of the isodesmic model, and alternative experimental validation of the conformation ensemble.

To address these points, we have performed additional analysis of the SAXS data and compared the results to an alternative measure of self-association. We have added an analysis in which use other thermodynamic models of self-association and compare the results to the SAXS data. We have also compared the model derived from the SAXS data to previously (but independently) measured CGMALS data. We tested the effect of the structural ensembles on the agreement with the SAXS data, both by subsampling the existing ensembles and by testing the agreement gained by using conformational ensembles compared to using individual (static) structures. The results from these analyses are consistent with the previous conclusions of the manuscript, and we thank the reviewers for the suggestions.

Reviewer #1 (Recommendations for the authors):I think the approach and results are well organized and presented. I don't have much to add. The only suggestion I have is to add some discussions regarding the biological implications of the rigid structures established from this work. For example, to what degree such rigidity is relevant to the phase separation behaviors of SPOP?

Following the suggestion from the reviewer, we have added results and discussion on the more biological implications. We have performed new molecular dynamics simulations of several disordered SPOP substrates to investigate the relationship between the spacing of binding motifs in SPOP substrates and the distances between substrate binding sites in SPOP oligomers. These results provide some functional context for the linear structures and spacing of the MATH domains in SPOP. Another potential function of rigidity has been explored in earlier work by Schmit et al. (https://doi.org/10.1021/jacs.9b10066), where it is proposed that the rigidity of SPOP oligomers is important to avoid a high entropic penalty upon stacking cross-bound oligomers in the dense phase. Our results are consistent with this model, and we have now included this in the Discussion section of the manuscript.

Reviewer #3 (Recommendations for the authors):The work could go further in more thoroughly testing each of the assumptions in light of the experimental data:Does the SAXS data preclude alternate association modes that would give rise to a significant fraction of non-linear structures? The introduction mentions the role of SPOP in condensate formation and one could imagine shorter oligomers associating in different ways than simply growing into single long chains.

We agree with the reviewer that, in the current work, we cannot strictly exclude that other non-linear models of self-association are consistent with the SAXS data. However, we show that the simple model of linear oligomers based only on the interfaces seen in crystal structures are consistent with the SAXS data including fitting a concentration-dependence consistent with an isodesmic model. Thus, we would argue that linear oligomers are the most plausible model based on this and other existing experimental evidence: 1. The only known interfaces for SPOP selfassociation are the BACK-BACK and BTB-BTB interfaces; 2. Removal or mutation of either the BACK-BACK or BTB-BTB interface results in abolishment of higher-order oligomerization; 3. Linear isodesmic self-association is consistent with both the SAXS and CG-MALS data; 4. Native mass spectrometry shows that higher-order oligomers are formed through the self-association of dimers, with every step of subunit-addition populated

(https://doi.org/10.15252/embj.201593169). To make this assumption clearer to the reader, we have added a paragraph discussing this in the Discussion section.

How would the fit to the SAXS data change if oligomer sizes are distributed differently, not following an isodesmic model?

We agree with the reviewer that it is useful to examine whether the SAXS data directly support the isodesmic model over other simple self-association models, independently of the previous existing evidence for the isodesmic model. To examine this problem further, we have now tested the agreement with the SAXS data modelling either (i) monodisperse oligomers or (ii) dimer-oligomer equilibria for a range of oligomer sizes; our results show that the isodesmic model gives better agreement with the SAXS data than any of these models. Additionally, it has previously been shown that CG-MALS experiments (covering a wider range of concentrations than our SAXS experiments) clearly exclude the possibility of smaller monodisperse oligomers (https://doi.org/10.15252/embj.201593169).

How does the sampling of the conformational ensembles affect the fits? One question would be how the results depend on the number of snapshots extracted from the simulations. Another question would be how consistent the generated domain-domain configurations are with the experimental data. The latter question is addressed in part by showing that a truncated ensemble where the MATH and BTB/BACK domains are in close contact does not lead to a good agreement. What about ensembles that only consist of (i) and (iii), or ensembles that consist only of (ii) and (iii)?

We agree with the reviewer that the manuscript would benefit from further analysis of how the agreement with the SAXS data is affected by the ensembles and the sampled conformations. To address the first point, we have tested how well single conformers drawn from the ensembles agree with the SAXS data. This analysis shows that a full ensemble is needed for optimal agreement with SAXS, but in line with the overall rigidity of oligomers, a small number of “well-selected” single structures can also give reasonable agreement with the SAXS data. To address the second point, we have added an analysis where we subsample the ensembles and assess the effect on the agreement with the SAXS data. The results from this analysis are consistent with the results from reweighting and restraining the simulations. However, we were not able to subsample ensembles with only a subset of the MATH configurations, as suggested by the reviewer, because the configurations of the MATH domains are not coordinated across different subunits, making it very unlikely to observe all MATH domains in an oligomer in the same configuration in a given frame. Addressing this problem in further detail would require extensive new simulations which we deem to be beyond the scope of this work.

Even though there is good agreement between the computational models and the SAXS data, further experimental validation of the proposed oligomers with different techniques may be warranted. While such experiments are likely beyond the scope of this work, some ideas about such experiments may be helpful.

We agree with the reviewer that further experimental validation would be interesting. We have added a direct comparison with CG-MALS data using data predicted from our SAXS-derived model of oligomerization.